# Coincidence of cholinergic pauses, dopaminergic activation and depolarisation of spiny projection neurons drives synaptic plasticity in the striatum

John N. J. Reynolds [1,5✉], Riccardo Avvisati [2], Paul D. Dodson [2], Simon D. Fisher [1], Manfred J. Oswald [1], Jeffery R. Wickens [1,3] & Yan-Feng Zhang [1,4,5✉]

Dopamine-dependent long-term plasticity is believed to be a cellular mechanism underlying reinforcement learning. In response to reward and reward-predicting cues, phasic dopamine activity potentiates the efficacy of corticostriatal synapses on spiny projection neurons (SPNs). Since phasic dopamine activity also encodes other behavioural variables, it is unclear how postsynaptic neurons identify which dopamine event is to induce long-term plasticity. Additionally, it is unknown how phasic dopamine released from arborised axons can potentiate targeted striatal synapses through volume transmission. To examine these questions we manipulated striatal cholinergic interneurons (ChIs) and dopamine neurons independently in two distinct in vivo paradigms. We report that long-term potentiation (LTP) at corticostriatal synapses with SPNs is dependent on the coincidence of pauses in ChIs and phasic dopamine activation, critically accompanied by SPN depolarisation. Thus, the ChI pause defines the time window for phasic dopamine to induce plasticity, while depolarisation of SPNs constrains the synapses eligible for plasticity.

[1] Department of Anatomy, University of Otago, School of Biomedical Sciences, Brain Health Research Centre, P.O. Box 913 Dunedin, New Zealand. [2] School of Physiology, Pharmacology & Neuroscience, University of Bristol, Bristol BS8 1TD, UK. [3] Okinawa Institute of Science and Technology, Okinawa 904-2234, Japan. [4] Department of Physiology, Anatomy & Genetics, University of Oxford, Oxford OX1 3PT, UK. [5] These authors contributed equally: John N. J. Reynolds, Yan-Feng Zhang. ✉email: john.reynolds@otago.ac.nz; yanfeng.zhang@dpag.ox.ac.uk

In the wild, animals need to take particular actions to maximise reward in a given situation. After obtaining a reward, or a sensory cue that predicts a reward, midbrain dopamine neurons briefly increase their firing rate in synchrony[1]. In reinforcement learning, phasic dopamine activity is thought to reinforce the actions that led to a reward to increase the chances of earning the same reward in future[2,3]. The cellular mechanism of reinforcement learning is believed to involve dopamine-dependent long-term plasticity, where increased dopamine levels enhance the efficacy of glutamatergic synapses between cortex and striatum[4–6]. Thus, striatal spiny projection neurons (SPNs) involved in executing a rewarded action will respond more effectively to subsequent presentations of the same cortical input pattern, in order to maximise future reward delivery. However, dopamine neurons not only increase their firing rate after reward and reward-predicting cues, but also increase to other encoded variables such as distance to reward[7,8], movement[9,10], and behavioural choices[11], although these variables may be encoded in a smaller discrete population of dopamine neurons[12]. Therefore, it is unclear how corticostriatal synapses might identify the appropriate dopamine signal from which to induce long-term plasticity during reinforcement learning.

In the current study, we tested the previously-proposed hypothesis[13–18] that the striatal tonically active neurons (TANs), likely to be ChIs, play a critical role in determining which dopamine signal is involved in modulating synaptic transmission. Although only 1% of striatal neurons are ChIs, they can regulate SPNs directly[19] and indirectly[20,21], and can modulate axonal dopamine release during coincident phasic dopamine activity[22–25]. ChIs exhibit excitation-pause-rebound multiphasic activity in response to reward and reward-indicating cues[26–28]. Critically, the pause phase coincides with phasic dopamine activity during learning[14,15]. Therefore, the ChI pause has been suggested to facilitate dopamine-dependent plasticity[13–18]. However, this hypothesis has never been tested in an intact brain in vivo due to the challenge of inducing synchronised pauses in sparsely distributed ChIs, aligned with phasic activity of dopamine neurons, within a physiologically meaningful timescale.

We recently showed that the firing of ChIs is entrained to fluctuations in excitatory input[29]. This observation enables us to test for the first time whether the ChI multiphasic response is involved in dopamine-dependent plasticity. Using the inverted striatal local field potential (iLFP) as a read-out of spontaneous and electrically evoked excitatory activity in our anaesthetised preparation[29], we are able to manipulate ChIs with cortical stimulation at the same time as we manipulate dopamine neurons. Here, we hypothesised that a temporal coincidence of a pause in the multiphasic activity of ChIs and phasic dopamine is required to induce long-term potentiation of corticostriatal synapses. Further, we propose that a marginal shift of the timing of phasic dopamine activity (a few hundred milliseconds) to overlap with excitation phases of ChI will switch the direction of long-term plasticity.

Another currently unresolved question in dopamine-dependent plasticity is how widespread changes in dopamine signalling in the striatum induce long-term plasticity only at those synapses involved in driving actions that lead to reward. Dopamine neurons have extensively arborised axons such that a single neuron can cover up to 5% of the striatum[30]. Furthermore, the volume transmission of dopamine in the striatum will further enhance the opportunity for synchronised phasic dopamine activity to elevate dopamine levels at synapses irrelevant to the rewarded action. One possible mechanism for constraining synaptic change is to limit plasticity to those postsynaptic neurons depolarised during the action. In slice preparations, depolarisation of the postsynaptic neurons is necessary to form long-term plasticity[31] and a conjunction of pre- and

postsynaptic activity, plus a reward signal (a three-factor plasticity rule) has been proposed[5]. However, the role of depolarisation has not been adequately characterised in vivo. Here, we further hypothesise that depolarisation of the postsynaptic SPNs is necessary for the induction of long-term plasticity, and the function is to constrain dopamine-dependent plasticity to the targeted synapses.

To test these hypotheses, we undertook two distinct in vivo experiments where we manipulated both midbrain dopamine neurons and putative ChIs (pChIs) so that phasic dopamine was present at different phases of the ChI multiphasic response. In addition, in the second set of experiments we also manipulated the membrane potential of the postsynaptic SPN by injecting positive current intracellularly, to determine the optimal timing of depolarisation for long-term plasticity. Our results suggest that the optimal conditions for induction of in vivo long-term potentiation at corticostriatal synapses on SPNs are a coincidence of phasic activity in dopamine neurons, pauses in ChIs, and depolarisation of postsynaptic SPNs.

## Results

Two experimental paradigms were used to investigate the integration and timing of glutamatergic, dopaminergic, and cholinergic signals in the striatum.

**The coincidence of ChI pause and phasic dopamine is required to induce LTP.** In the first experimental paradigm, we made single-unit extracellular recordings of SPNs in urethane-anaesthetised rats (Fig. 1a, b, Supplementary Fig. 1). Contralateral cortical electrical test pulses were applied to elicit spikes in SPNs, and a change in the probability of cortically-evoked spikes was used to quantify corticostriatal plasticity. In addition to evoking spikes in SPNs, the cortical stimulation also entrained the firing of striatal ChIs (Fig. 1c), with the pause indicated by the 'receding phase' and maximal firing during the 'rising phase' of the iLFP, as described previously[29,32] (Fig. 1c).

Dopamine neurons were activated phasically by a physiologically relevant stimulation, a light flash to the contralateral eye when the superior colliculus (SC) was disinhibited by local injection of the GABA antagonist bicuculline (BIC; Fig. 1a). During collicular disinhibition, a visual stimulus drives phasic activity in dopamine neurons at about 110 ms after a light flash[33,34] (Supplementary Fig. 2), as well as contributing to a slower response (latency ~200 ms) in pChIs (Fig. 1c). Thus, by pairing cortical stimulation with disinhibited visual stimulation at varying intervals, we were able to activate phasic dopamine activity at the timing of either a pause or excitation of pChIs, respectively (Fig. 1c).

To determine whether the coincidence of the ChI pause and phasic dopamine is required for potentiation in SPNs, we first acquired baseline cortical responses by applying cortical test stimulation alone for 10 min at 0.2 Hz (see Fig. 2b for paradigm). Then, to induce synaptic plasticity, cortical stimulation was paired with light stimulation. After 5 min of pairing, the SC was disinhibited locally by BIC and then pairing continued for another 10 min. The combination of these two manipulations either drove phasic dopamine activity to coincide with a pChI pause (the 'Match' group) or with pChI excitation ('Mismatch' group). Then, spike activity of SPNs in response to cortical test stimulation alone was continued for at least another 30 mins. We found that the spike activity of SPNs induced by cortical stimulation (total number of spikes at short-latency; < 30 ms) remained potentiated only when the period of pChI pauses and phasic dopamine was coincident ('Match' group, Fig. 2a left and 2b). In contrast, depression of spike activity occurred in the 'Mismatch' group (Fig. 2a right and 2b), in which dopamine was temporally separated from the pChI pause.

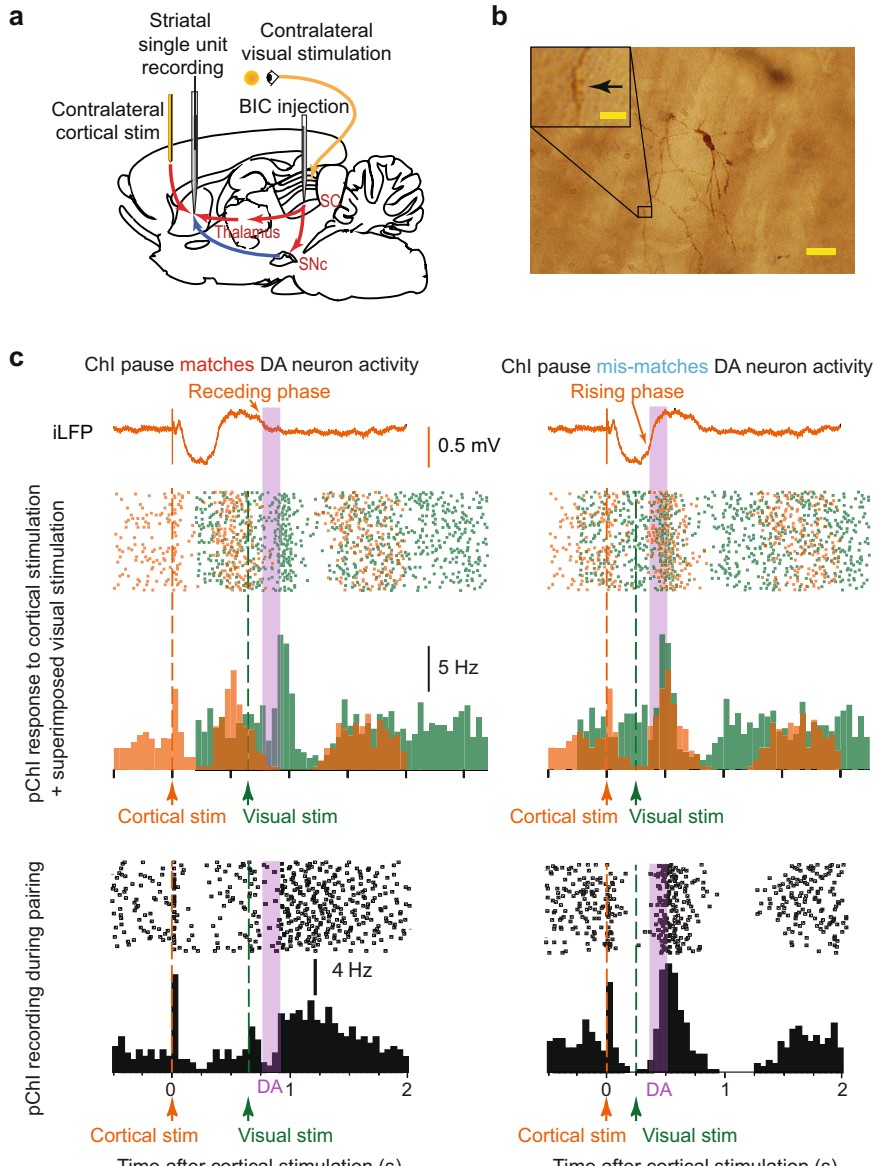

**Fig. 1 Experimental paradigm to induce coincident ChI pause, dopamine activation and depolarisation. a** Cortical stimulation was applied to elicit spike activity in SPNs and to entrain the activity of pChIs. Contralateral visual stimulation was applied with BIC injected into the SC to activate dopamine neurons and depolarise SPNs. **b** A representative example from 11 labelled SPNs with spine arrowed in inset (Scale bar 20 μm; inset 5 μm). **c** *Upper,* Cortical stimulation alone entrained the firing pattern of a pChI (orange raster and histogram). The excitation phase of the pChI occurs at the rising phase of the iLFP (orange trace), and the pause phase occurs at receding phases of the iLFP. Superimposed is the pChI response to the visual stimulation alone (green) with the SC disinhibited, and the likely period of released dopamine following visual activation indicated (purple). The visual stimulation is aligned to the timing used for pairing in the lower panels to illustrate the effect of the alignment. Because the excitation induced by the visual stimulation is very slow, it did not interfere with the pause induced by the critical stimulation. *Lower,* pairing cortical stimulation and visual stimulation at different intervals can drive phasic dopamine activity to coincide with either a ChI pause (*left*) or excitation phase (*right*). Note, due to the difficulties of recording pChI in vivo, three different pChIs were recorded to demonstrate how cortical stimulation alone (*upper, orange*), or visual stimulation alone (*upper, green*), or pairing of two stimuli (*lower*) regulate the firing pattern of pChIs, respectively.

We further tested whether the pChI pause alone can induce potentiation. In the control group ('Control'), light flashes were only paired with cortical stimulation for the 5 min prior to bicuculline injection and not subsequently, so pChI pauses were entrained by the cortical stimulation but dopamine neurons were not phasically-activated. We found a similar depression in the 'Control' group as the 'Mismatch' group (Fig. 2b). Thus, pChI pauses elicited by cortical stimulation were insufficient to induce potentiation without phasic dopamine. Therefore, potentiation of corticostriatal responses

only emerged when phasic dopamine was coincident with the pChI pause.

However, using this particular experimental paradigm we were unable to test our second hypothesis regarding the necessity of postsynaptic depolarisation for corticostriatal plasticity in vivo. In this preparation, SPNs were depolarised by the light stimulation via the tecto-thalamo-striatal glutamatergic pathway at a similar time as dopamine neurons were activated (Fig. 1a)[35,36]. Therefore, we used a second experimental paradigm to allow us to

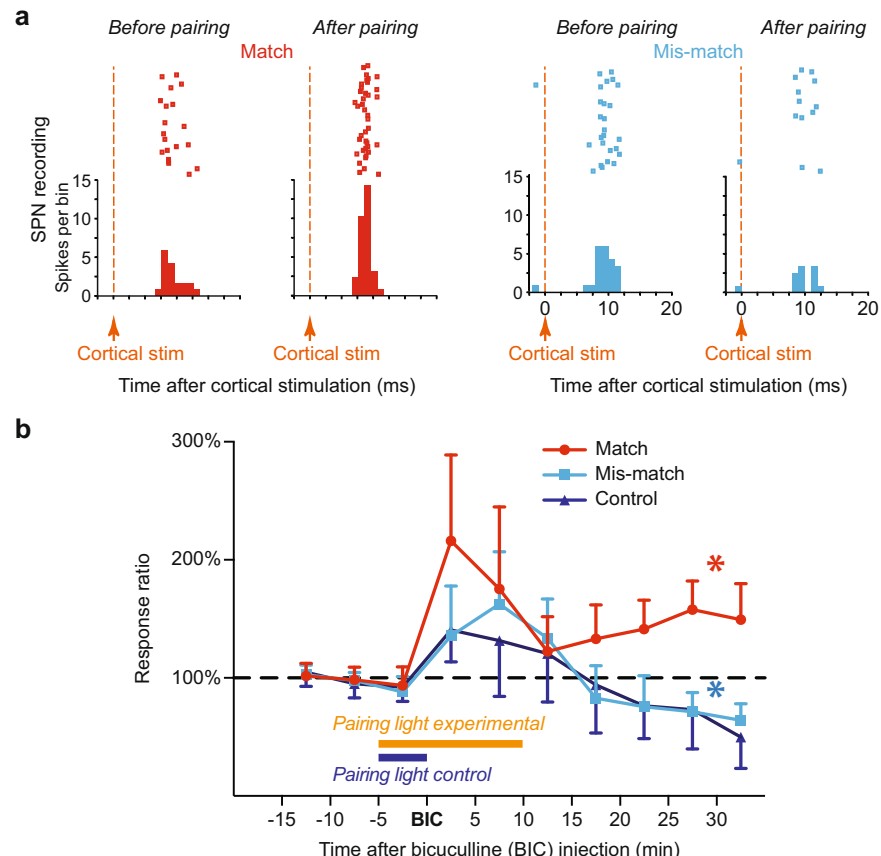

**Fig. 2 SPN spikes induced by cortical stimulation were potentiated by coincident ChI pause, dopamine activation and depolarisation. a** Cortical stimulation induced spike activity in SPNs. A representative potentiation (left) and depression (right) in SPNs 30 min after match or mismatch pairings, respectively. **b** Spike activity in SPNs (mean ± S.E.M.) was potentiated when the light-induced dopamine signal was coincident with ChI pauses (red, $p = 0.0227$), but not when coincident with excitation of ChIs (light blue, $p = 0.0349$) or when phasic dopamine was absent (dark blue—no light flashes after BIC, $p = 0.2219$). $N = 6$ neurons in each group, *$p < 0.05$ (two-tailed paired $t$ test; averaged last 10 min compared to baseline).

manipulate SPN depolarisation independently of SNc dopamine cell activation (Fig. 3a, b).

**The depolarisation of postsynaptic SPNs is also required for potentiation.** We tested whether depolarisation of postsynaptic SPNs is required for LTP in vivo by using a paradigm with which we have previously demonstrated an association between synaptic plasticity and positive reinforcement[4]. In these experiments, we applied a contralateral electrical test stimulation to simulate cortical input, and directly activated dopamine neurons with electrical stimulation in the midbrain. Intracellular recordings were made to depolarise the postsynaptic SPNs and measure the postsynaptic potential (PSP).

In our previous study[4] we found that dopamine-dependent long-term potentiation (LTP) of synapses on recorded SPNs was induced when the neurons were depolarised during the SPN 'down state' using intracellular current injection, and when dopamine neurons were activated simultaneously. However, according to our recent investigation of the factors underlying the firing activity of ChIs[29], the down state of SPNs not only corresponds to a period of minimum excitatory cortical input to the striatum[37], but also to the time when ChIs exhibit a pause (Fig. 3c). Therefore, serendipitously, the LTP in our previous study was elicited by activation of dopamine neurons and the depolarisation of SPNs during the period of a pChI pause. However, the necessity of each of the factors contributing to LTP were not further elucidated in that study.

Here we used a similar paradigm to determine the temporal requirements for activation of dopamine neurons, the pChI pause and depolarisation of SPNs for the induction of corticostriatal LTP. We first successfully replicated our previous finding of LTP induction by applying the same combination of dopamine neuron stimulation and depolarisation in the down state, which we now know corresponds to the ChI pause, in naïve animals (Match: $17.2 \pm 10.2\%$ at +20 min, $N = 7$ animals; Fig. 3c–e).

We then tested if the presence of the ChI pause was necessary for the induction of LTP. In contrast, we found that when SPN depolarisation and dopamine input were applied at the time that ChIs are most excited, during the spontaneous SPN up state, that no significant change in synaptic efficacy resulted (Mismatch: $-1.0 \pm 5.1\%$ at 20 min, $N = 7$ animals; Fig. 3e). Notably, the difference in plasticity resulting in the Match and the Mismatch group was not due to differences in the level of membrane depolarisation achieved, since current injection in both groups was set to just exceed the threshold for action potential firing (see Methods). Thus, these results agree with the extracellular experiments in demonstrating a need for dopamine activation to coincide with the pause in ChIs to induce lasting potentiation.

We further tested whether the combination of dopamine and the pause in ChIs, or the combination of SPN depolarisation and the pause in pChIs, is effective in inducing LTP. We found that either dopamine stimulation alone (SNc stim only: $-14.0 \pm 4.4\%$ at +20 min, $N = 5$ animals; $p < 0.001$ in comparison to match group) or depolarisation of SPNs alone (Depol only: $-15.8 \pm 6.1\%$ at

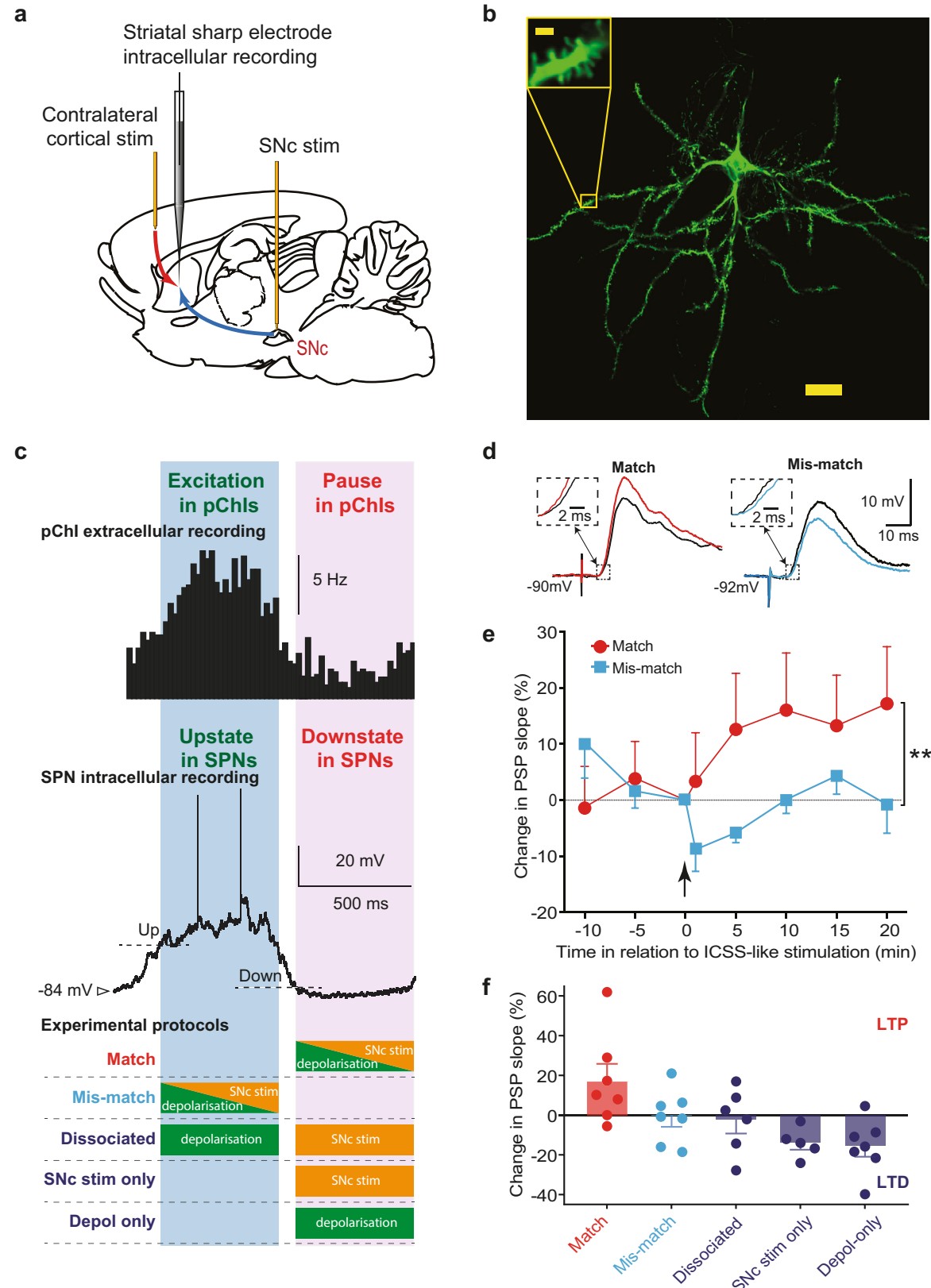

20 min, $N = 7$ animals; $p < 0.01$ in comparison to the match group) applied during the pChI pause period, induced LTD.

Finally, we tested whether coincident SPN depolarisation and dopamine signalling are required for LTP induction. We temporally separated the depolarisation and dopamine stimulation. We applied the depolarisation immediately following SPN up state initiation, at the excitation phase of pChIs, and dopamine input was applied in the following down state corresponding to the pause phase in ChIs. We found that temporally dissociating these components led to no change in synaptic efficacy (Dissociated: $-2.7 \pm 6.6\%$ at 20 min, $N = 6$ animals; $p < 0.01$ in comparison to match group, Fig. 3c, f).

**Fig. 3 Coincidence of ChI pause, dopamine activation and depolarisation potentiated corticostriatal SPN postsynaptic potentials (PSPs). a** Cortical stimulation induced PSPs in SPNs. Dopamine neurons were activated by electrical stimulation and depolarisation was induced by intracellular current injection. **b** A representative example from 11 labelled SPNs (Scale bar 20 μm; inset 2 μm). **c** Electrical stimulation of the SNc was set to match (purple shade) or mismatch (blue shade) the ChI pause, determined from the striatal iLFP. **d** Example corticostriatal PSPs potentiated (red) or depressed (light blue) in slope (expanded in insets) and in amplitude 20 min after baseline (black traces) when SNc stimulation matched or mismatched the ChI pause, respectively. **e** Group average effect on PSPs of SNc stimulation matched ($N = 7$) vs mismatched ($N = 7$); **$p < 0.01$ (Mean ± S.E.M.; unpaired $t$ test at 20 min, $p = 0.0025$). **f** One-way ANOVA with Brown-Forsythe and Welch ANOVA post hoc tests to compare Match group (which induced LTP on average) to all other groups (where no change or LTD was induced): Match vs. Mismatch $p = 0.008$, Match vs. Dissociated ($N = 6$) $p = 0.008$, Match vs. SNc stim only ($N = 5$) $p = 0.0005$, Match vs. Depol-only ($N = 7$) $p = 0.0010$.

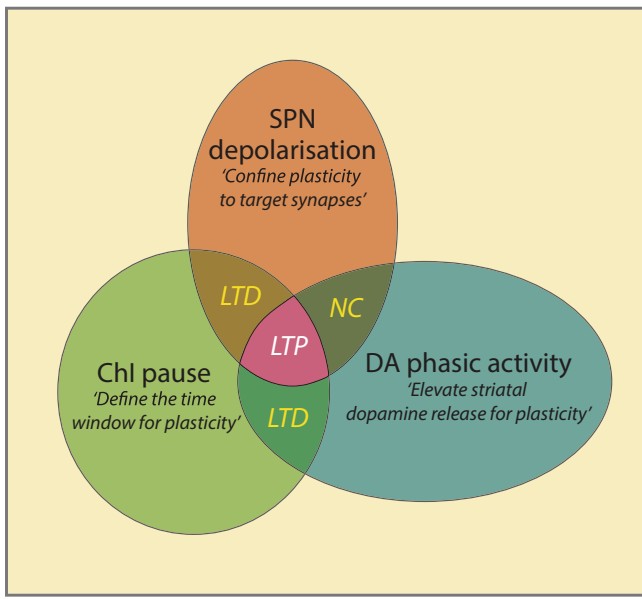

**Fig. 4 A model of corticostriatal synaptic plasticity.** Corticostriatal LTP (pink) is induced when SPNs are depolarised (brown) in a striatal area where a ChI pause response (green) is coincident with a phasic dopamine signal (blue). Other conditions will result in LTD or no change (NC).

## Discussion

Despite differences in recording techniques and measures of plasticity, results from both extracellular and intracellular recordings were in agreement, that temporal coincidence of phasic dopamine, pChI pause and SPN depolarisation is required for the induction of corticostriatal LTP in vivo. No change or synaptic depression is found if these signals occur during the excitation phase of the pChIs, if dopamine signalling or SPN depolarisation occur alone during the pause phase, or if they are separated in time around the pChI pause (Fig. 4).

Our results suggest that the ChI pause may define a time window for phasic dopamine to induce long-term synaptic potentiation. We here entrained pChIs to pause with cortical electrical stimulation[29]. Compared to optogenetic manipulations[20,21,23,38], which depress a subpopulation of ChIs to reduce spike activity, activation of the cortex can ensure that most of the pChIs pause in synchrony[29,39] to mimic the synchronised pChI pause observed in behavioural animals[26,27]. In addition, the pChI pause entrained by cortical stimulation is naturally flanked by endogenous excitations, which may themselves encode reward-relevant information[40,41]. Thalamic input in particular has a prominent role in inducing multiphasic ChI responses during learning[42,43], and is harnessed in Experiment 1 to provide the critical depolarisation required for plasticity, through the effect of the light flash activating the disinhibited tecto-thalamic pathway[35,36]. Further work is required to combine cortical and thalamic inputs to induce ChI multiphasic activity to fully mimic neuronal activities in freely moving animals and to test their role in

synaptic plasticity and learning. In addition, a GABAergic input from the ventral tegmental area (VTA) may also contribute to the formation of the ChI pause in the ventral striatum[44], indicating acetylcholine and dopamine system may interplay in a different manner in the ventral striatum than dorsal striatum.

We further demonstrated that changes in pChI activity without phasic dopamine do not induce long-term potentiation at corticostriatal synapses (control group in experiment 1, depol-only group in experiment 2), consistent with others' findings[38]. Thus, because phasic dopamine activity reduces at late stages of learning whereas the ChI pause remains intact, the resulting effect of the interplay between ChI activity and dopamine afferents at the striatal level may be dependent on the stage of learning. In addition, manipulating dopamine neurons to fire phasically can only induce synaptic potentiation if it coincides with the pChI pause but not the excitation phase, indicating that the ChI pause is the time window for synaptic potentiation.

We here also addressed how phasic dopamine contributes to synaptic plasticity without using dopamine receptor antagonists, which undesirably also block the effects of tonic dopamine. Indeed, a previous ex vivo study showed that tonic activation of dopamine receptors can reverse the direction of long-term plasticity induced by cortical stimulation[45]. To avoid blocking the tonic effects of dopamine, our control experiments instead altered the timing of phasic dopamine or omitted dopamine neuron stimulation altogether. We found that even shifting the timing of phasic dopamine activity marginally by a few hundred milliseconds is powerful enough to reverse the direction of the long-term plasticity. Also, when phasic activation of dopamine neurons was omitted, LTD or no change of synaptic efficacy was found. It should be noted that our study focused on mimicking the cell body activity in ChIs and dopamine neurons, as revealed earlier[15]. However, the release profile of dopamine may not precisely mirror the firing pattern of dopamine neurons due to the potential axonal release driven by ChIs[23,24], and should be addressed in future work. Overall, these experiments indicate that the LTP we observed is dependent on phasic dopamine.

Our results also provide further support for the requirement for depolarisation of the target postsynaptic SPNs for long-term potentiation in vivo[46] and is consistent with previous ex vivo observations[31]. This suggests that depolarisation of postsynaptic SPNs may play a critical role in constraining long-term potentiation to synapses on neurons engaged in the action of retrieving rewards. While SPNs in the direct pathway and indirect pathway express distinct sets of dopamine and muscarinic receptors, these neurons may be potentiated to different degrees during learning, which is worth addressing in future studies. In summary, our study provides empirical data supporting the important temporal gating of corticostriatal synaptic plasticity by a physiologically-induced pause response in ChIs and depolarisation of the postsynaptic SPNs.

## Methods

All procedures in this study were conducted in accordance with approvals granted by the University of Otago Animal Ethics Committee or in accordance with the UK

animals scientific procedures act, 1986 with approval of the University of Bristol ethics committee. A total of 136 male Long-Evan rats for extracellular recording were used, yielding 258 putative spiny projection neurons, and 18 were successfully recorded for the full plasticity recording protocol. The intracellular recording was performed with 105 male Wistar rats, yielding 32 spiny projection neurons recorded for the plasticity protocol.

**Surgery.** Male Long–Evans rats (250–450 g) or Wistar rats (280– 50 g) were anaesthetised with urethane (1.4–1.9 g/kg i.p.; Biolab Ltd., Auckland, New Zealand). During recording, the level of anaesthesia was monitored by continuous observation of the bandpass filtered EEG signal (0.01–500 Hz). Supplementary urethane was administered via an intraperitoneal catheter at any sign of EEG desynchronisation, indicating a lessening of depth of anaesthesia. The head was fixed in a stereotaxic frame (Narishige, Japan) and core temperature maintained above 36 °C by a homoeothermic blanket and rectal probe (TR-100, Fine Science Tools). All wounds and pressure points were infiltrated with a long-acting local anaesthetic (Bupivacaine, 0.5%).

For monitoring of the electroencephalogram (EEG), a hole was drilled in the skull above the left posterior cortex, and a silver wire electrode placed against the dura overlying the cortex and fixed in placed with dental cement. A flap of bone overlying the cortex was removed to provide access to the recording site in the left medial striatum, and a "well" of dental cement fashioned around the perimeter of the hole. All coordinates are given in millimetres in relation to Bregma and the midline.

**Electrical stimulation.** To implant a stimulating electrode into the medial agranular motor cortex, a round piece of skull overlying the right hemisphere (centred AP + 2.0 to +2.7 mm and ML −1.6 to −2.0 mm to Bregma) was removed. A concentric (extracellular experiments; Rhodes NEW-100 × 10 mm, USA) or parallel-contact (intracellular experiments, locally manufactured) stimulating electrode was implanted in the medial agranular motor cortex to a depth of 1.6 to 2.4 mm. Stimulating electrodes were connected to constant current electrical stimulators (Isolator-10, Axon Instruments Inc.) Stimulus pulses applied to the cortex were biphasic (0.1–0.2 Hz, 0.1 ms, 300 to 990 μA). For experiments requiring substantia nigra stimulation, the medial contact of a parallel-contact bipolar stimulating electrode was implanted at interaural coordinates AP + 3.4 to +3.6; ML + 1.6; DV 2.1 to 2.3. Substantia nigra stimulation consisted of 50 biphasic pulses (0.5 ms duration) applied at 100 Hz (average current applied for each group 500 to 990 uA).

**Visual stimulation.** In the extracellular recording experiments, visual stimuli (10 ms duration, 0.2 Hz) were delivered by a white LED (1500 mcd) that was placed 1–2 cm directly in front of the right eye of the animal. The left eye was covered. LED and electrical stimulating electrodes were connected to constant current electrical stimulators (Isolator-10, Axon Instruments Inc.).

**Bicuculline injections.** The drug-filled pipettes were lowered to 4.0–4.2 mm from the brain surface into the deep layers of the superior colliculus (AP −6.5/ ML + 1.5 mm), and either supported by the IVM micromanipulator or secured with dental cement. Bicuculline (0.01% in saline, 250 nl) was injected into the superior colliculus at a rate of 400 nl/min.

**Extracellular recording.** Extracellular single-unit recordings were made using 5–15 MΩ micropipettes. Electrodes were filled with 1 M NaCl solution with 2% neurobiotin (SP1120, Vector). Only stable striatal neurons with wide average spike waveform (>1.1 ms), and slow spontaneous firing rate (<0.1 Hz), e.g. Supplementary Fig. 2, were included. Neurons with train spike activity typical of low threshold-spiking (LTS) neurons were also excluded from this study. The spike rate of the recorded SPNs during 30 ms following each cortical stimulation was used as an indication of the strength of corticostriatal synapses. Recordings were made via either a headstage (model HS-2A) connected to an Axoprobe-1A microelectrode amplifier (Axon Instruments Inc California, USA), or a headstage (NL 100 Neurolog) connected to a preamp (NL104), an amplifier (NL106) and a filter (NL125). Signals were amplified and band-pass filtered (0.1–10,000 Hz). All waveform data were digitised at 50 kHz by an A-D interface (1401 Micro 2, CED, UK), and acquired using SPIKE2 software (v6 or v7, CED).

Dopamine neurons were extracellularly recorded and juxtacellularly labelled (AP −5(±0.3) ML + 2(±0.3) DV −7/−8) using glass electrodes as described above. Single-unit data were acquired with an ELC-01MX headstage (NPI electronic), amplified 1000 times, and bandpass filtered at 300 to 5000 Hz (DPA-2FS filter). Depth of anaesthesia was monitored using the electrocorticogram (amplified 2000 times and bandpass filtered at 300 to 1500 Hz) from a stainless-steel screw implanted at AP +2, ML +2. Data were digitised at 25 kHz using a CED Power 1401 mkII.

**Intracellular recording.** Intracellular recordings were made using 35–130 MΩ micropipettes with 1 M K-acetate internal solution, in some cases containing 3–4% biocytin. For intracellular recording, only stable neurons with a membrane

potential more negative than −60 mV that displayed characteristic spontaneous fluctuations in membrane potential (>10 mV amplitude) and action potential firing were included in this study. Current–voltage relations were obtained by injecting hyperpolarising and depolarising current pulses through the micropipette, using an Axoclamp-2B amplifier (Molecular Devices) configured in current-clamp mode. Membrane potential fluctuations were recorded for periods of at least a minute after the cell had stabilised following impalement and at regular intervals of >15 min. All waveform data were digitised at 10 kHz by a Digidata 1200B or a Digidata 1322A (Molecular Devices), displayed using pClamp 8 software (Molecular Devices).

**Extracellular recording experimental protocol.** After a stable single-unit recording was obtained from a putative spiny projection neuron, cortical stimulation (0.2 Hz) was applied throughout the recording, and the short latency (<30 ms) spike response was measured as the strength of the corticostriatal synaptic input. During the recording, firstly, a baseline of 10 min of spike activity was recorded. The cortical stimulation was then paired with light stimulation for another 5 min. Bicuculline was then injected locally to the deep layers of the superior colliculus. Visual stimulation was paired for another 10 min with the cortical stimulation. In the match group, visual stimulation was applied when the iLFP started to decrease, approximately 500 to 600 ms after cortical stimulation. In the mismatch group, visual stimulation was applied when the iLFP started to increase, approximately 250 ms after cortical stimulation. Therefore, the estimated light-induced phasic dopamine activity occurred either during the ChI pause (Match group) or excitation (Mismatch group, Fig. 1c). In the control group, the visual stimulation was not applied after BIC injection. After pairing, the recording of spike responses to cortical stimuli continued for at least another 30 min.

The peristimulus histograms (PSTH) of putative ChIs and SPN spikes were plotted using the cortical stimulation (Figs. 1c, 2a) or the trough of the iLFP (Fig. 3c) as the triggers. Each PSTH represents 60 sweeps of recording and the bin sizes were 50, 1, and 20 ms in Figs. 1c, 2a, and 3c, respectively.

**Intracellular recording experimental protocol.** After a stable impalement was obtained from a spiny projection neuron, a baseline of 20 min of postsynaptic potentials (PSPs) was recorded before the plasticity-inducing protocol. Postsynaptic potentials were always elicited from the Down state. The transitions between membrane potential "states" were detected online using a locally-constructed functional clamp[47], and this used to trigger data acquisition and substantia nigra electrical stimulation, where relevant, early in the ensuing Down or Up state. A delay of 600 ms was imposed between intracellular current injection and substantia nigra stimulation in the dissociated group to ensure that the depolarisation was induced in the up state, and the substantia nigra stimulation occurred primarily in the ensuing down state. Intracellular current injection, when included in the protocol, was set 0.2 nA higher than the threshold for eliciting continuous action potential firing (range 0.5–2.0 nA). Membrane potential fluctuations and current–voltage relations were assessed before and after the plasticity-inducing protocol to ensure that changes in PSPs were not associated with changes in membrane characteristics. The experimental protocols used are illustrated in Fig. 3c.

**Histology.** At the end of extracellular recording, the putative spiny neurons were actively filled with neurobiotin by a juxtacellular filling protocol[48]. Briefly, the target cells were "driven" to spike by applying positive current through the recording pipette (up to 12 nA, 250 ms on–off pulses) for up to 15 min. At the end of intracellular recording, putative spiny neurons were filled intracellularly with biocytin by applying depolarising current pulses (0.8–1.5 nA; 3 Hz; 10–15 min) via the recording micropipette.

Vibratome sections (50–60 μm) were processed using standard histological procedures[49] and labelled cells were identified by light or fluorescent microscopy. In the extracellular recording experiments, 8 neurons, and in the intracellular recording experiments, 14 neurons, were recovered and verified histologically as spiny projection neurons. Juxtacellularly labelled dopamine neurons were identified as previously described (Dodson et al., 2016) using Cy3-conjugated streptavidin (1:500, Sigma-Aldrich GEPA43001), chicken anti-Tyrosine Hydroxylase primary (1:500, Abcam AB76442) and Brilliant Violet 421-conjugated anti-chicken secondary antibodies (1:500, Jackson ImmunoResearch 703-675-155)[50].

Positions of the substantia nigra electrodes for intracellular recordings were determined in unstained or cresyl-violet stained sections to be within 500 μm of the dopamine cell layer of the substantia nigra pars compacta. There were no systematic differences in electrode positions between groups. In groups where SN stimulation was not applied, electrodes were still fitted during surgery to control for the release of dopamine that may accompany acute electrode implantation[51].

**Data analysis.** Data were analysed offline using built-in functions of Axograph 4.9 software and SPIKE2 v6 or v7. Statistical tests on data from single cells as well as on group data were performed in SPSS and Prism.

**Reporting summary**. Further information on research design is available in the Nature Research Reporting Summary linked to this article.

## Date availability

Raw data for Figs. 1 and 2 are available if required. Source data for figures are provided with the paper. Source data are provided with this paper.

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

## Acknowledgements

Our thanks to Annabel Kean and Koreen Clements for technical assistance, and to Jan Schulz, Peter Redgrave and Gordon Arbuthnott for helpful discussions that have shaped this manuscript. This work was supported by grants from the PhD scholarship, Department of Anatomy, University Otago (to Y-FZ), Marsden Fund of the Royal

Society of NZ (to JNJR, and JRW/JNJR), Lottery Health Research (JNJR/JRW) and the Neurological Foundation of New Zealand (MJO/JNJR).

## Author contributions

Conceptualisation, Y.-F.Z., J.R.W., and J.N.J.R.; Data curation, Y.-F.Z. and J.N.J.R.; Investigation, Y.-F.Z., extracellular recordings, J.N.J.R. intracellular recordings: R.A., P.D.D. dopamine neuron recordings; Formal analysis, Y.-F.Z., extracellular recordings, J.N.J.R intracellular recordings, R.A., P.D.D. dopamine neuron recordings; Methodology, Y.-F. Z., J.R.W., M.J.O. and J.N.J.R.; Funding acquisition and resources, J.R.W. and J.N.J.R.; Writing—original draft, Y.-F. Z. and J.N.J.R.; Writing—review & editing, Y.-F.Z., S.D.F., M.J.O., J.R.W., R.A., P.D.D., and J.N.J.R.; Supervision, J.N.J.R.

## Competing interests

The authors declare no competing interests.
