## [Peer Review File · Nature Communications]

Coincidence of cholinergic pauses, dopaminergic activation and depolarisation of spiny projection neurons drives synaptic plasticity in the striatumREVIEWER COMMENTS

Reviewer #1 (Remarks to the Author):

Coincidence of cholinergic pauses, dopaminergic activation and depolarization drives synaptic plasticity in the striatum.

This interesting manuscript presents novel data using in vivo recording approaches to investigate how plasticity of the cortical excitatory input to striatal projection neurons is controlled by the interplay between the activity of tonically active striatal neurons, which are presumed to be cholinergic, and the phasic activation of Dopamine (DA) input to the striatum. The authors nicely introduce this area of research, its importance, and the logic behind their experimental paradigms.

In experiment one, the authors used an interesting technique to drive phasic DA release which has been previously studied, where the dis-inhibition of the superior colliculus has revealed the powerful driving of substantia nigra DA neurons, which in turn drives the phasic release of DA in the striatum (although the authors have not demonstrated that this occurs in their preparations). The authors then paired the presumed phasic release of DA with excitatory input to the striatum (to drive both cholinergic neurons and projection neurons). The authors note that they were able to entrain activity, presumably the UP-down states apparent under the anaesthetic conditions used, to produced periods of phasic activation of putative cholinergic neurons, which was followed by a pause, and then a rebound increase in action potential output. The authors have published this finding, but as illustrated in the current manuscript this is far from clear, in the examples shown the (Figure 1C), the upper raster and PSTH show a nice early excitation and pause, while in the lower raster and PSTH, the kinetics of this sequence are quite distinct, where the pause is very brief, followed by a period of firing and a second pause. Clearly therefore across preparations this behaviour is not stereotyped, making it difficult to ascertain the robustness of any further temporal pairing with DA neuron activation. For example, in both illustrated cases, the presumed time of DA release does not correspond with the pause or peak of rebound excitation, but in both illustrated cases corresponds with the rising phase of the rebound excitation. As the authors have stated that a large fraction of presumed cholinergic neurons of the striatum are synchronized by their procedures, it seems that the illustrated examples do not show what the authors are indicating, perhaps this is an incorrect interpretation on my behalf, and so this issue requires more further analysis to be made clear to the reader. In this regard I was also surprised by the PSTHs of MSN activity (Figure 1d), I take from the authors methods that this represents events per bin, do these neurons fire at a rate of > 500 Hz, or is this a product of how the authors have illustrated their data. It is also unclear how these PSTHs are translated into the results presented in Figure 1e, was the area under the PSTH calculated or the total number of APs? With the caveats attached to the relative timing of presumed DA release and the firing rate of putative cholinergic neurons it is unclear to this reviewer if the presented results are consistent with the idea of matched and mismatched cases.

The experimental paradigm presented in Figure 2 is also of interest, but description of the results is too brief to allow full evaluation. The presented data suggests that robust UP – down states are apparent, which occur spontaneously (Figure 2c, not d). I presume in this case the cortical stimulation was set so as not to entrain this activity, but to produce small amplitude EPSPs in MSN neurons. The timing of events appears more robust in this paradigm, and it would be beneficial to see multiple cycles of the UP-down states and analysis for the sample of recordings reported. In this case, electrical stimulation of the DA neurons is paired with excitatory input and a spontaneous UP- state in the striatum, then it appears that a long term change, acceleration in the rise-time of EPSPs is apparent, although this is a small change, and not possible to observe from the illustrated EPSPs (Figure 2d), which rather focus on an amplitude change. The presented results however do show a differential impact that is UP-down state dependent, which is of great interest.

In summary, the manuscript contains interesting data. This reviewer could not judge the robustness of observations, within the context of the current presentation. I would suggest that more robust analysis will help the reader understand the relative timing of activity in the system, and the meaningful of the results.

Reviewer #2 (Remarks to the Author):

Cholinergic interneurons (ChIs) of the striatum play a role in detecting motivationally relevant stimuli. Both ChIs and midbrain dopamine (DA) neurons projecting to the striatum emit coincident signals in response to rewarding events and it has been suggested that the dynamic interplay between these signals is crucial for reward-guided learning. This manuscript deals with an interesting issue : what are the conditions required for the development of synaptic plasticity (i.e., the primary mechanism for learning and memory) in striatal output neurons (the so-called spiny projection neurons or SPNs) ? In this study, Zhang and colleagues provide in vivo experimental data showing that these conditions required a coincidence of phasic activity in DA neurons, pauses in ChI tonic firing and depolarization of SPNs. It therefore appears that integration and timing of dopaminergic, cholinergic, and glutamatergic signals in the striatum represent a powerful triumvirate involved in long-term potentiation at corticostriatal synapses on output neurons.

There are two novel and exciting results: (1) Although the mechanisms underlying the pause in ChI tonic firing are still debated, the present study indicates that pause generation can be driven by a reduction of their excitatory inputs; (2) Temporal coincidence of phasic DA activation and pauses in ChIs has long been proposed as a putative mechanism for the induction of synaptic plasticity in the striatum and the authors add a third critical component: depolarization of postsynaptic output neurons in the striatum. These results allow to specify how striatal DA transmission is constrained in time and space. As the

authors state: "the ChI pause defines the time window for phasic DA to induce plasticity, while depolarization of SPNs constrains the synapses eligible for plasticity". These findings have important implications for striatal plasticity and the interaction between striatal neuromodulatory systems in pathological states. The manuscript is concise - a bit too much on some aspects -, well written, and the data as a whole are interesting and clearly presented. There are some difficulties and limitations that need to be addressed, however, as discussed below.

Major

(1) For a non-specialist reader, the Introduction may seem a bit confusing at times. It is not clear if the main working hypothesis is new or comes from the literature. It has been proposed for several years that synchrony between DA bursts and ChI pauses may represent a mechanism required to keep the DA release effective onto postsynaptic targets in the striatum. As Zhang et al. acknowledge, the contribution of the Bergman lab in the field is indisputable, suggesting early that a change in cholinergic tone provides a time window for DA-induced modification of synaptic weights (Morris et al. 2004). It would be desirable to better emphasize that the hypothesis had already been put forward and that experimental evidence was lacking. I understand that the authors have made an effort to do so in the Introduction, but it needs to be revised to remove any ambiguity.

(2) The authors used cortical stimulation to drive ChI activity in anesthetized rats that mimics the pause in ChI firing seen after reward-predicting stimuli in behaving animals. They demonstrate that synchrony between DA activation and ChI pause are two key factors for the induction of synaptic plasticity in the striatal output. In the early stages of learning, during stimulus-action-reward association, this might be the way it works. However, an idea anchored in the literature considers that DA activation is reduced as learned behavioral responses become more habitual, whereas ChIs continue to respond. This suggests that the interplay between the effects of the ChIs and those of the DA afferents at the striatal level are dependent on the stage of learning. What can be the function of the ChI pause in the absence of a DA burst in routinized behavior? This would be worth mentioning in the Discussion.

(3) Afferents from intralaminar thalamic nuclei arborize over large regions of the striatum and are well positioned to produce a synchronized reduction in excitatory input that target preferentially ChIs. Also, inactivation of the CM-Pf in monkeys results in a loss of the ChI pause (Matsumoto et al. 2001) and findings suggest that the thalamic input to ChIs exerts state or contextual control over striatal plasticity (e.g., Bradfield et al. 2013). Based on their experimental approach, Zhang et al. mainly consider a cortical influence, but the thalamic influence can be predominant. Can you please refer to it in the Discussion?

(4) The mechanisms responsible for pause generation in ChIs have long been investigated and different scenarios have been considered, though they are difficult to reconcile. The present study clearly shows that a reduction in excitatory inputs to ChIs is a factor. Interestingly, long-range GABAergic inputs to ChIs from the ventral tegmental area (i.e., direct inhibitory influence) have been reported to pause ChI firing

in the nucleus accumbens (Brown et al., 2012). Although their study did not target the ventral striatum, can the authors speculate (if space allows) on different mechanisms between dorsal and ventral parts of the striatum ?

(5) The impact of ACh and DA on corticostriatal synaptic plasticity is perhaps more subtle depending on the type of SPN. In indirect pathway SPNs that express D2 DA receptors, DA depresses excitability and ACh increases it through activation of M1 muscarinic receptors. In direct pathway SPNs that express D1 DA receptors, the situation appears to be more nuanced by the coexpression of M1 and M4 muscarinic receptors. Zhang et al. did not address the issue of whether the type of depolarized SPN can be important for the development of synaptic plasticity in striatal output pathways. Their experimental approach does not allow to tackle this issue, but it should be mentioned for future research.

Minor

(1) In their previous paper (Zhang et al. 2018), the authors refer to the tonically active neurons they recorded in the striatum as "putative cholinergic neurons". In these times of suspected heterogeneity of "basic" neuronal types in the striatum, I would do the same in the present paper.

(2) Typo. Introduction, line 45 "to maximize" vs. line 54 "to maximise"

(3) It would be welcome to briefly explain what is an "inverted local field potential" and how useful it is for the experiments described in the present study.

(4) The authors manipulated DA neurons and ChIs so that phasic changes in DA activity occurred at different phases of the ChI response. This is a nice experimental design to test the hypothesis that a particular ChI response component (the pause) is coupled to the DA burst to induce plasticity. However, they mention a point in the Introduction that hides the interest of this design: "Here, we hypothesised that the multiphasic activity of ChIs defines a time window for phasic DA to induce long-term potentiation on corticostriatal synapses" (Introduction, lines 78-80). This sentence needs to be rewritten to place emphasis on the temporal resolution of the approach.

(5) Although the paper appears under-illustrated, the Fig.3 is a bit too simplistic and not very enlightening. I would recommend to exclude it or to improve it - if possible - to offer a graphical synthesis of the findings.

Reviewer #3 (Remarks to the Author):

This is an interesting manuscript aiming to show in vivo that dopamine dependent plasticity in striatum is gated by pauses in cholinergic activity in concert with depolarization of MSNs. In previous work they showed that depolarization from down state and simultaneous DA activation was necessary for LTP (Reynolds et al., 2001). While observed in vitro, confirming in vivo is best. Here, the authors present basically two types of experiments, Figure 1 and Figure 2, suggesting that indeed this may be the case. Overall I like the paper, but there are some key things that need to be shown and accounted for.

Figure 1: Important to show that disinhibition of SC and visual stimulation leads to DA neuron activation and subsequent dopamine release in striatum in the present data set. At this point, it is inferred from prior works. As DA release has been shown to be regulated via differing mechanisms than solely somatic firing, it is important to show.

How was time of DA phasic activity estimated?

Disinhibition of SC and visual stimulation also leads to thalamostriatal activation. Should one show that ChI pause and thalamic activation are not driving changes in SPN firing to make the stated claim?

Figure 1e – not clear if this is the average of last two data points vs average of baseline?

Figure 2: It seems that both questions – matching of ChI pause/dopamine stim and SPN depolarization – could have been answered using the second experimental paradigm. Not completely clear why Figure 1 was not performed using Figure 2 set up- but may be missing something.

Indirect stimulation of SNc via visual stim and SC disinhibition introduces confounds in the induction of plasticity as SC projections to other regions such as thalamus may contribute/modulate the induction of plasticity. This may be more in the interpretation, but should be made more clear.

REVIEWER COMMENTS

Reviewer #1 (Remarks to the Author):

Coincidence of cholinergic pauses, dopaminergic activation and depolarisation drives synaptic plasticity in the striatum.

This interesting manuscript presents novel data using in vivo recording approaches to investigate how plasticity of the cortical excitatory input to striatal projection neurons is controlled by the interplay between the activity of tonically active striatal neurons, which are presumed to be cholinergic, and the phasic activation of Dopamine (DA) input to the striatum. The authors nicely introduce this area of research, its importance, and the logic behind their experimental paradigms.

In experiment one, the authors used an interesting technique to drive phasic DA release which has been previously studied, where the dis-inhibition of the superior colliculus has revealed the powerful driving of substantia nigra DA neurons, which in turn drives the phasic release of DA in the striatum (although the authors have not demonstrated that this occurs in their preparations). The authors then paired the presumed phasic release of DA with excitatory input to the striatum (to drive both cholinergic neurons and projection neurons). The authors note that they were able to entrain activity, presumably the UP-down states apparent under the anaesthetic conditions used, to produced periods of phasic activation of putative cholinergic neurons, which was followed by a pause, and then a rebound increase in action potential output. The authors have published this finding, but as illustrated in the current manuscript this is far from clear, in the examples shown the (Figure 1C), the upper raster and PSTH show a nice early excitation and pause, while in the lower raster and PSTH, the kinetics of this sequence are quite distinct, where the pause is very brief, followed by a period of firing and a second pause. Clearly therefore across preparations this behaviour is not stereotyped, making it difficult to ascertain the robustness of any further temporal pairing with DA neuron activation. For example, in both illustrated cases, the presumed time of DA release does not correspond with the pause or peak of rebound excitation, but in both illustrated cases corresponds with the rising phase of the rebound excitation. As the authors have stated that a large fraction of presumed cholinergic neurons of the striatum are synchronised by their procedures, it seems that the illustrated examples do not show what the authors are indicating, perhaps this is an incorrect interpretation on my behalf, and so this issue requires more further analysis to be made clear to the reader.

We thank the reviewer for pointing out that a better description of how cortical and visual stimulation regulates pChIs is needed.

In experiment 1, we used cortical stimulation to entrain the pChIs. Visual stimulation was applied to drive phasic dopamine cell body activity to coincide with either the excitation or the pause phase of pChIs.

However, visual stimulation (with BIC injected to SC) used here also regulates the firing pattern of pChIs via the tecto-thalamo-striatal pathway (Schulz *et al*, 2011). When applied alone, visual stimulation can induce a slow excitation (~200 ms latency and 200 ms long) followed by a 2s long pause (Schulz *et al.*, 2011).

Therefore, when cortical and visual stimulation were applied in the pairing protocol, both of these will contribute to the firing pattern of pChIs and therefore the firing pattern of pChIs look different. However, because the ChI response to visual stimulation is much slower, we

can use it to drive phasic dopamine activity to coincide with either the pause or excitation driven by cortical stimulation.

To clarify this, we have now added a new **Supplementary Fig 1** to demonstrate when cortical stimulation was applied alone after BIC injection, the firing rate of pChI reaches a maximum when iLFP increases and a trough when iLFP decreases, as we described in Zhang et al., 2018 Neuron. We further show that visual stimulation induces a slow excitation (latency > 200 ms) in ChIs, which is followed by a long slow pause in pChIs. This visual stimulation induced response is consistent with our previous finding (Schulz et al., 2011, Goldberg & Reynolds, 2011; see Figure below). Finally, we demonstrate how cortical and visual stimulations regulate the firing pattern of pChIs when they are applied at different intervals, as in **Fig 1**.

When SC is disinhibited, visual stimulation induces a slow excitation and a long pause in a pChI in an anaesthetised rat. (Adapted from Goldberg & Reynolds 2011)

In this regard I was also surprised by the PSTHs of MSN activity (Figure 1d), I take from the authors methods that this represents events per bin, do these neurons fire at a rate of > 500 Hz, or is this a product of how the authors have illustrated their data. It is also unclear how these PSTHs are translated into the results presented in Figure 1e, was the area under the PSTH calculated or the total number of APs? With the caveats attached to the relative timing of presumed DA release and the firing rate of putative cholinergic neurons it is unclear to this reviewer if the presented results are consistent with the idea of matched and mismatched cases.

We have now addressed this apparent high spike frequency in SPNs by adding a new supplementary figure. The un-usual 'high firing rate' of SPNs is a product of the narrow bin width of the PSTH (2 ms). These narrow bins (2 ms) were used to demonstrate the variation of spike latencies following the cortical stimulation before and after the pairing protocol. To further clarify this, we have changed the y-axis and units from 'firing rate (Hz)' to 'spikes per bin' in **figure 1d**.

However, it is worth noting that these neurons rarely spike spontaneously, i.e. the firing rate of these neurons is less than 0.1Hz. We have now added a new Supplementary Fig 3 as an example. This is a continuous recording from a SPN. As showed in the figure, this SPN only

responded to the cortical stimulation but has no spontaneous spike activity during a 10 s recording.

We further address this in the methods section that all neurons included in this study have a low spontaneous firing rate that is less than 0.1 Hz (Line 322- line 323).

We also clarified in the main text that the total number of action potentials immediately following the cortical stimulation (< 30 ms) contributed to Fig 1E in the main text (Line 139- line 141).

The experimental paradigm presented in Figure 2 is also of interest, but description of the results is too brief to allow full evaluation. The presented data suggests that robust UP – down states are apparent, which occur spontaneously (Figure 2c, not d). I presume in this case the cortical stimulation was set so as not to entrain this activity, but to produce small amplitude EPSPs in MSN neurons. The timing of events appears more robust in this paradigm, and it would be beneficial to see multiple cycles of the UP-down states and analysis for the sample of recordings reported. In this case, electrical stimulation of the DA neurons is paired with excitatory input and a spontaneous UP- state in the striatum, then it appears that a long term change, acceleration in the rise-time of EPSPs is apparent, although this is a small change, and not possible to observe from the illustrated EPSPs (Figure 2d), which rather focus on an amplitude change. The presented results however do show a differential impact that is UP-down state dependent, which is of great interest.

We thank the reviewer and agree the results are of great interest. We agree it would be useful to show multiple cycles of the UP-down states. However, we, unfortunately, did not have the continuous recordings of the membrane potentials of SPNs because of the nature of episodic recording of intracellular signals. Each episode of recordings was triggered by a threshold discriminator when the membrane potential transition to either Up or Down state (Reynolds and Wickens, 2003, *Journal of Neuroscience Methods*). To verify the triggering system worked properly during recordings, we monitored the membrane potential with an oscilloscope. However, this signal was unfortunately not recorded.

To address the illustration of EPSPs, we have now modified Figure 2d to add an enlarged time base inset and demonstrate how the slopes of the EPSPs changed for LTP (increase) and LTD (decrease).

In summary, the manuscript contains interesting data. This reviewer could not judge the robustness of observations, within the context of the current presentation. I would suggest that more robust analysis will help the reader understand the relative timing of activity in the system, and the meaningful of the results.

We thank the reviewer and agree the manuscript is of interest, and hope our new figures and additional experimental data will help clarify the manuscript.

Reviewer #2 (Remarks to the Author):

Cholinergic interneurons (ChIs) of the striatum play a role in detecting motivationally relevant stimuli. Both ChIs and midbrain dopamine (DA) neurons projecting to the striatum emit coincident signals in response to rewarding events and it has been suggested that the

dynamic interplay between these signals is crucial for reward-guided learning. This manuscript deals with an interesting issue : what are the conditions required for the development of synaptic plasticity (i.e., the primary mechanism for learning and memory) in striatal output neurons (the so-called spiny projection neurons or SPNs) ? In this study, Zhang and colleagues provide in vivo experimental data showing that these conditions required a coincidence of phasic activity in DA neurons, pauses in Ch1 tonic firing and depolarisation of SPNs. It therefore appears that integration and timing of dopaminergic, cholinergic, and glutamatergic signals in the striatum represent a powerful triumvirate involved in long-term potentiation at corticostriatal synapses on output neurons.

There are two novel and exciting results: (1) Although the mechanisms underlying the pause in Ch1 tonic firing are still debated, the present study indicates that pause generation can be driven by a reduction of their excitatory inputs; (2) Temporal coincidence of phasic DA activation and pauses in Ch1s has long been proposed as a putative mechanism for the induction of synaptic plasticity in the striatum and the authors add a third critical component: depolarisation of postsynaptic output neurons in the striatum. These results allow to specify how striatal DA transmission is constrained in time and space. As the authors state: “the Ch1 pause defines the time window for phasic DA to induce plasticity, while depolarisation of SPNs constrains the synapses eligible for plasticity”. These findings have important implications for striatal plasticity and the interaction between striatal neuromodulatory systems in pathological states. The manuscript is concise - a bit too much on some aspects -, well written, and the data as a whole are interesting and clearly presented. There are some difficulties and limitations that need to be addressed, however, as discussed below.

Major

(1) For a non-specialist reader, the Introduction may seem a bit confusing at times. It is not clear if the main working hypothesis is new or comes from the literature. It has been proposed for several years that synchrony between DA bursts and Ch1 pauses may represent a mechanism required to keep the DA release effective onto postsynaptic targets in the striatum. As Zhang et al. acknowledge, the contribution of the Bergman lab in the field is indisputable, suggesting early that a change in cholinergic tone provides a time window for DA-induced modification of synaptic weights (Morris et al. 2004). It would be desirable to better emphasise that the hypothesis had already been put forward and that experimental evidence was lacking. I understand that the authors have made an effort to do so in the Introduction, but it needs to be revised to remove any ambiguity.

We have now emphasised this point in the main text. We further clarify that the Bergman group hypothesised that Ch1s and dopamine neurons co-activate dopamine-dependent learning as early as 2004, and we here provided the experimental evidence, as stated in line 61- line 63 as below.

- ‘In the current study, we tested the previously-proposed hypothesis¹³⁻¹⁸ that the striatal tonically active neurons (TANs), likely to be Ch1s, play a critical role in determining which dopamine signal is involved in modulating synaptic transmission.’

(2) The authors used cortical stimulation to drive Ch1 activity in anaesthetised rats that mimics the pause in Ch1 firing seen after reward-predicting stimuli in behaving animals. They demonstrate that synchrony between DA activation and Ch1 pause are two key factors for

the induction of synaptic plasticity in the striatal output. In the early stages of learning, during stimulus-action-reward association, this might be the way it works. However, an idea anchored in the literature considers that DA activation is reduced as learned behavioral responses become more habitual, whereas ChIs continue to respond. This suggests that the interplay between the effects of the ChIs and those of the DA afferents at the striatal level are dependent on the stage of learning. What can be the function of the ChI pause in the absence of a DA burst in routinised behavior ? This would be worth mentioning in the Discussion.

We agree this is an essential question in learning! In the control group of experiment 1 and 'Depol only' group in experiments, we showed the pause in ChIs alone would not induce LTP but rather a trend towards LTD. These results are consistent with Lee et al., 2016 where they showed that the ChI pause alone could not facilitate LTP or reinforcement learning. We propose that when the phasic dopamine signal decreases, the pause response in ChIs may gradually decrease the efficacy of synaptic transmission until animals need to re-learn the task. The phasic dopamine signal will then appear again to facilitate synaptic efficacy.

We have added this argument to the Discussion in line 234- line 237 as below

- 'Thus, because phasic dopamine activity reduces at late stages of learning whereas the ChI pause remains intact, the resulting effect of the interplay between ChI activity and dopamine afferents at the striatal level may be dependent on the stage of learning.'

(3) Afferents from intralaminar thalamic nuclei arborise over large regions of the striatum and are well positioned to produce a synchronised reduction in excitatory input that target preferentially ChIs. Also, inactivation of the CM-Pf in monkeys results in a loss of the ChI pause (Matsumoto et al. 2001) and findings suggest that the thalamic input to ChIs exerts state or contextual control over striatal plasticity (e.g., Bradfield et al. 2013). Based on their experimental approach, Zhang et al. mainly consider a cortical influence, but the thalamic influence can be predominant. Can you please refer to it in the Discussion ?

In this study, we have used cortical stimulation to entrain the firing pattern of ChIs. However, the thalamic input no doubt plays a critical role in regulating ChI firing patterns. In our earlier study (Zhang et al., 2018), we proposed that excitatory inputs may regulate the multiphasic activity in ChIs, and these excitatory inputs could come from both cortex and thalamus, which is unfortunately not fully understood yet. We have now added this to our Discussion as suggested by the reviewer in line 221 -line 227 as below

- 'Thalamic input in particular has a prominent role in inducing multiphasic ChI responses during learning^{42, 43}, and is harnessed in Experiment 1 to provide the critical depolarisation required for plasticity, through the effect of the light flash activating the disinhibited tecto-thalamic pathway^{35, 36}. Further work is required to combine cortical and thalamic inputs to induce ChI multiphasic activity to fully mimic neuronal activities in freely moving animals and to test their role in synaptic plasticity and learning.'

(4) The mechanisms responsible for pause generation in ChIs have long been investigated and different scenarios have been considered, though they are difficult to reconcile. The present study clearly shows that a reduction in excitatory inputs to ChIs is a factor. Interestingly, long-range GABAergic inputs to ChIs from the ventral tegmental area (i.e., direct inhibitory influence) have been reported to pause ChI firing in the nucleus accumbens

(Brown et al., 2012). Although their study did not target the ventral striatum, can the authors speculate (if space allows) on different mechanisms between dorsal and ventral parts of the striatum ?

We thank the reviewer for mentioning the GABA role in ChI pauses. We agree that any input that potentially contributes to the firing pattern of ChIs should have contributions to dopamine-dependent long-term plasticity. We have now added that GABAergic inputs to ChIs may also contribute to the dopamine-dependent synaptic plasticity in the striatum in line 227- line 230 as below.

- 'In addition, a GABAergic input from the ventral tegmental area (VTA) may also contribute to the formation of the ChI pause in the ventral striatum⁴⁴, indicating acetylcholine and dopamine system may interplay in a different manner in the ventral striatum than dorsal striatum.'

(5) The impact of ACh and DA on corticostriatal synaptic plasticity is perhaps more subtle depending on the type of SPN. In indirect pathway SPNs that express D2 DA receptors, DA depresses excitability and ACh increases it through activation of M1 muscarinic receptors. In direct pathway SPNs that express D1 DA receptors, the situation appears to be more nuanced by the coexpression of M1 and M4 muscarinic receptors. Zhang et al. did not address the issue of whether the type of depolarised SPN can be important for the development of synaptic plasticity in striatal output pathways. Their experimental approach does not allow to tackle this issue, but it should be mentioned for future research.

We thank the reviewer for this suggestion! We have added this to our Discussion in line 260 – line 262 as below.

- 'While SPNs in the direct pathway and indirect pathway express distinct sets of dopamine and muscarinic receptors, these neurons may be potentiated to different degrees during learning, which is worth addressing in future studies.'

Minor

(1) In their previous paper (Zhang et al. 2018), the authors refer to the tonically active neurons they recorded in the striatum as "putative cholinergic neurons". In these times of suspected heterogeneity of "basic" neuronal types in the striatum, I would do the same in the present paper.

We have changed ChIs in our experiment to pChIs throughout.

(2) Typo. Introduction, line 45 "to maximise" vs. line 54 "to maximise"

Fixed.

(3) It would be welcome to briefly explain what is an "inverted local field potential" and how useful it is for the experiments described in the present study.

We have added Supplementary Fig 1 to address how the firing pattern of pChIs correlates with the 'inverted local field potential (iLFP)'.

(4) The authors manipulated DA neurons and ChIs so that phasic changes in DA activity occurred at different phases of the ChI response. This is a nice experimental design to test the hypothesis that a particular ChI response component (the pause) is coupled to the DA

burst to induce plasticity. However, they mention a point in the Introduction that hides the interest of this design: “Here, we hypothesised that the multiphasic activity of ChIs defines a time window for phasic DA to induce long-term potentiation on corticostriatal synapses” (Introduction, lines 78-80). This sentence needs to be rewritten to place emphasis on the temporal resolution of the approach.

We have re-written the sentence to ‘Here, we hypothesised that a temporal coincidence of a pause in the multiphasic activity of ChIs and phasic dopamine is required to induce long-term potentiation of corticostriatal synapses. Further, we propose that a marginal shift of the timing of phasic dopamine activity (a few hundred milliseconds) to overlap with excitation phases of ChI will switch the direction of long-term plasticity.’ Line 78 – line 82.

(5) Although the paper appears under-illustrated, the Fig.3 is a bit too simplistic and not very enlightening. I would recommend to exclude it or to improve it - if possible - to offer a graphical synthesis of the findings.

We have modified figure 3 by adding ‘define the time window for plasticity’ to ChIs and ‘confine plasticity to target synapses’ to SPNs.

Reviewer #3 (Remarks to the Author):

This is an interesting manuscript aiming to show in vivo that dopamine dependent plasticity in striatum is gated by pauses in cholinergic activity in concert with depolarisation of MSNs. In previous work they showed that depolarisation from down state and simultaneous DA activation was necessary for LTP (Reynolds et al., 2001). While observed in vitro, confirming in vivo is best. Here, the authors present basically two types of experiments, Figure 1 and Figure 2, suggesting that indeed this may be the case. Overall I like the paper, but there are some key things that need to be shown and accounted for.

Figure 1: Important to show that disinhibition of SC and visual stimulation leads to DA neuron activation and subsequent dopamine release in striatum in the present data set. At this point, it is inferred from prior works. As DA release has been shown to be regulated via differing mechanisms than solely somatic firing, it is important to show. How was time of DA phasic activity estimated?

We thank for the reviewer for pointing this out. We have been using 110ms as the latency of the phasic dopamine activity driven by visual stimulation which was first reported by Dommett et al. in 2005 (included here). We used this number because we closely collaborated with Pete Redgrave’s group, and we used a very similar experimental setup as Dommett et al., 2005.

Disinhibition of collicular deep layers induced phasic visual responses locally and in DA neurons. Initially, raster displays and peri-stimulus histograms show that collicular neurons and a simultaneously recorded DA neuron were unresponsive to a regular (0.5 Hz) light flash (vertical dotted line) (top graphs). After a collicular microinjection of bicuculline, both local neurons and the DA neuron were excited at short latency by visual stimulation (bottom graphs). (from Dommett et al., 2005)

However, as required by the reviewer, we have successfully replicated this finding with new single-unit recording experiments and, for the first time, use juxtacellular labelling to identify the neurons as dopaminergic. These experiments are shown in a new Supplementary Fig 2. In our hands, we also find the latency to the increased firing rate of dopamine neurons is just over 100 ms, and it lasts for ~150 ms.

We agree that dopamine release could also be driven by synchronised ChI activity (Threlfell et al., 2012, *Neuron*, Kosillo, Zhang et al., 2016, *Cerebral Cortex*), and it would be very interesting to understand how ChIs and spike activity of dopamine neurons drive dopamine release in the striatum. However, the profile of dopamine release is beyond the scope of this study. In the current study, we aimed to mimic the observation of cell body activity recorded in ChIs and dopamine neurons (Morris et al., 2004). Therefore, even if the initial excitation and/or the rebound phases of ChI multiphasic activity drives axonal dopamine release, it will not change the conclusions.

In addition, when we only apply cortical stimulation in the 'control group' of experiment 1, and when depolarisation of SPNs coincides with the pause in ChIs (depol only group' in experiment 2), we found LTD. Therefore, we are confident that even if the excitatory input, induced by cortical stimulation in this study, can drive axonal dopamine release via ChIs, this dopamine signal is not sufficient to drive LTP. We have added this argument to our discussion and will follow this lead in our future studies.

Disinhibition of SC and visual stimulation also leads to thalamostriatal activation. Should one show that ChI pause and thalamic activation are not driving changes in SPN firing to make the stated claim?

We thank the reviewer for this question. The disinhibition of SC and visual stimulation indeed will activate the thalamostriatal pathway to depolarise SPNs, as we showed previously (Schulz et al., 2011). In experiment 1, we used visual stimulation to activate dopamine neurons which also activates the thalamostriatal pathway, and unfortunately, there is no perfect control we could perform to separate these two pathways in this paradigm. We indeed agree with the reviewer that the depolarisation of SPNs induced by the activation of the thalamostriatal pathway will affect the long-term plasticity in the SPNs. To test this, we performed the second experiment to identify the role of depolarisation in SPNs in the long-term plasticity of SPNs.

Figure 1e – not clear if this is the average of last two data points vs average of baseline?

Yes, we compared the average of the last two data points vs the average of baseline. This is now clarified in the figure legend.

Figure 2: It seems that both questions – matching of ChI pause/dopamine stim and SPN depolarisation – could have been answered using the second experimental paradigm. Not completely clear why Figure 1 was not performed using Figure 2 set up- but may be missing something.

The reason we performed experiment 1 is to test whether activation of dopamine neurons with a physiologically meaningful (visual) stimulation, such as experienced by behaving animals, can induce long-term plasticity in SPNs. However, to activate dopamine neurons with visual stimulation under anaesthesia, we need to disinhibit SC by a local injection of BIC to allow the visual information to reach the dopamine neurons through the tectonigral pathway. Unfortunately, we found that injection of BIC during the very sensitive intracellular experiments caused instability in the recordings leading to loss of the cell. The successful rate of stable and long-lasting intracellular recordings with intracranial drug (BIC) injection is extremely low. Therefore, we have to choose to use extracellular recordings to increase the success rate. However, that does mean that we lose the opportunity to record and control the membrane potential in the recorded neurons.

However, we believe that the concurrence between the results from two distinct experimental paradigms make our conclusion more robust, and more physiologically meaningful.

Indirect stimulation of SNc via visual stim and SC disinhibition introduces confounds in the induction of plasticity as SC projections to other regions such as thalamus may contribute/modulate the induction of plasticity. This may be more in the interpretation, but should be made more clear.

Thank you for the useful advice! We have added this point to the discussion in line 221 – 227 as below.

- ‘Thalamic input in particular has a prominent role in inducing multiphasic ChI responses during learning^{42, 43}, and is harnessed in Experiment 1 to provide the critical depolarisation required for plasticity, through the effect of the light flash activating the disinhibited tecto-thalamic pathway^{35, 36}. Further work is required to combine cortical and thalamic inputs to induce ChI multiphasic activity to fully mimic neuronal activities in freely moving animals and to test their role in synaptic plasticity and learning.’

REVIEWER COMMENTS

Reviewer #1 (Remarks to the Author):

I found the manuscript little changed. I am still confused by the authors brevity of description and illustration. It seems to me that Sup fig 1 is necessary for understanding and should be moved to the main text. I find it hard to follow the below section, as it is so inelegantly written, and requires multiple reads to extract any meaning. How is the neuron depolarized during the down state? By the injection of current, by the evoked synaptic activity? The authors need to make this point clear. If it is by current injection is the same DC level achieved in the match and mis-matched states? As with the majority of the manuscript the authors had an opportunity to expand the text to increase readability but have chosen not too.

170 In our previous study⁴ we found dopamine-dependent long-term potentiation (LTP) on the
171 recorded SPNs when they were depolarised during the SPN 'down state' and dopamine neurons
172 were activated simultaneously. However, according to our latest study²⁹, the down-state of
173 SPNs not only corresponds to a period of minimum excitatory cortical input to the striatum³⁷,
174 but also to the time when ChIs exhibit a pause (Fig. 2c). Therefore, serendipitously, the LTP
175 in our previous study was elicited by activation of dopamine neurons and depolarised SPNs
176 during a pChI pause, but the factors contributing

Reviewer #2 (Remarks to the Author):

This paper has been improved by adding more analyses and clarification. The authors have carefully revised their manuscript and well addressed the issues I raised.

I have two last comments:

1. Appropriateness of the title (I didn't notice that before): The study shows that a coincidence of phasic activity in DA neurons, pause in striatal ChI firing and depolarisation of SPNs is required. Would it not be better to mention in the title "GABAergic output neuron depolarisation" rather than just "depolarisation" ? I know, this makes the title a bit cumbersome and I leave it to the authors to decide.

2. Figure 3: Although I appreciate the authors' effort to improve this figure as a graphical synthesis of their view about corticostriatal synaptic plasticity, I would suggest a few modifications in this Venn-like diagram:

(1) Move "SPN" inside the brown bubble

(2) I wonder if it is possible to insert a general tag information about the functional significance of « DA phasic activity » ?

I have no further comments and I congratulate the authors on this interesting work.

P. Apicella

Reviewer #3 (Remarks to the Author):

The author has nicely addressed my minor concerns. This is a very nice addition to the literature, and the authors should be commended.

We thank the three reviewers for their efforts to review our manuscript and their suggestions to improve it. Here, we detail how we have responded to their comments,

REVIEWER COMMENTS

Reviewer #1 (Remarks to the Author):

I am still confused by the authors brevity of description and illustration. It seems to me that Sup fig 1 is necessary for understanding and should be moved to the main text.

In the new version, we have merged half of Fig. 1 and supplementary figure 1 into a new Fig.1, and shifted the remainder of Fig. 1 to a new Fig. 2. Thus, the manuscript now contains four main figures and 2 supplementary figures.

How is the neuron depolarized during the down state? By the injection of current, by the evoked synaptic activity? The authors need to make this point clear. If it is by current injection is the same DC level achieved in the match and mis-matched states?

We emphasise at the end of the Introduction and in the section the Reviewer quotes below that depolarisation is achieved by intracellular current injection. In addition, later in the results section we clarify the Reviewer's point by adding:

Notably, the difference in plasticity resulting in the Match and the Mismatch group was not due to differences in the level of membrane depolarisation achieved, since current injection in both groups was set to just exceed the threshold for action potential firing (see Methods).

I find it hard to follow the below section, as it is so inelegantly written, and requires multiple reads to extract any meaning.....

17 In our previous study⁴ we found dopamine-dependent long-term potentiation (LTP) on the 171 recorded SPNs when they were depolarised during the SPN 'down state' and dopamine neurons

172 were activated simultaneously. However, according to our latest study²⁹, the down-state of 173 SPNs not only corresponds to a period of minimum excitatory cortical input to the striatum³⁷, 174 but also to the time when ChIs exhibit a pause (Fig. 2c). Therefore, serendipitously, the LTP 175 in our previous study was elicited by activation of dopamine neurons and depolarised SPNs 176 during a pChI pause, but the factors contributing

As suggested by reviewer 1, we have modified the section below to:

In our previous study⁴ we found that dopamine-dependent long-term potentiation (LTP) of synapses on recorded SPNs was induced when the neurons were depolarised during the SPN 'down state' using intracellular current injection, and when dopamine neurons were activated simultaneously. However, according to our recent investigation of the factors underlying the firing activity of ChIs²⁹, the down-state of SPNs not only corresponds to a period of minimum excitatory cortical input to the striatum³⁷, but also to the time when ChIs exhibit a pause (**Fig. 3c**). Therefore, serendipitously, the LTP in our previous study was elicited by activation of dopamine neurons and the depolarisation of SPNs during the period of a pChI pause. However, the necessity of each of the factors contributing to LTP were not further elucidated in that study.

We have also made various minor changes and corrections throughout as highlighted, to improve readability.

Reviewer #2 (Remarks to the Author):

This paper has been improved by adding more analyses and clarification. The authors have carefully revised their manuscript and well addressed the issues I raised.

I have two last comments:

1. Appropriateness of the title (I didn't notice that before): The study shows that a coincidence of phasic activity in DA neurons, pause in striatal Ch1 firing and depolarisation of SPNs is required. Would it not be better to mention in the title "GABAergic output neuron depolarisation" rather than just "depolarisation" ? I know, this makes the title a bit cumbersome and I leave it to the authors to decide.

Thank you for the suggestion. We have thus changed the title to 'Coincidence of cholinergic pauses, dopaminergic activation and depolarisation *of spiny projection neurons* drives synaptic plasticity in the striatum'.

2. Figure 3: Although I appreciate the authors' effort to improve this figure as a graphical synthesis of their view about corticostriatal synaptic plasticity, I would suggest a few modifications in this Venn-like diagram:

(1) Move "SPN" inside the brown bubble

(2) I wonder if it is possible to insert a general tag information about the functional significance of « DA phasic activity » ?

We have moved 'SPNs' to the brown bubble as suggested, and also added 'Elevate striatal dopamine level for plasticity' to the DA phasic activity.

I have no further comments and I congratulate the authors on this interesting work.

We thank the Reviewer for their comments.

Reviewer #3 (Remarks to the Author):

The author has nicely addressed my minor concerns. This is a very nice addition to the literature, and the authors should be commended.

We thank the Reviewer for their positive comments.

REVIEWERS' COMMENTS

Reviewer #1 (Remarks to the Author):

The authors have addressed my concerns.

Reviewer #2 (Remarks to the Author):

The authors have taken into account my suggestions. I have no further comments on the manuscript.

Reviewer #3 (Remarks to the Author):

No further comments

REVIEWERS' COMMENTS

Reviewer #1 (Remarks to the Author):

The authors have addressed my concerns.

We thank the Reviewer for their positive comments.

Reviewer #2 (Remarks to the Author):

The authors have taken into account my suggestions. I have no further comments on the manuscript.

We thank the Reviewer for their positive comments.

Reviewer #3 (Remarks to the Author):

No further comments

We thank the Reviewer for their positive comments.